**Title: Dimensionality and flexibility of learning in biological recurrent neural networks**

**Scientific Question:** Does full-rank gradient descent accurately describe the dynamics of synaptic plasticity in biological recurrent neural networks?

**Core Collaborators:**
**Senior:**
**Blake Richards (Mila/ McGill University)**: will develop new theory, debate topic with other senior adversarial collaborators, write position paper, attend CCN
**Claudia Clopath (Imperial College London)**: will develop new theory, debate topic with other senior adversarial collaborators, write position paper
**Rui P Costa (University of Bristol)**: will develop new theory, debate topic with other senior adversarial collaborators, write position paper
**Wolfgang Maass (Graz University of Technology)**: will develop new theory and models, debate topic with other senior adversarial collaborators, write position paper

**Junior:**
**Luke Prince (Mila/ McGill University)**: will develop new theory, incorporate feedback from CCN community, organize kickoff GAC workshop, write position paper, and attend CCN
**Arna Ghosh, Roy Eyono (Mila/ McGill University)**: will develop new theory, incorporate feedback from CCN community, organize kickoff GAC workshop, write position paper
**Franz Scherr, Martin Pernull (Graz University of Technology):** will develop new theory and models, including the study of learning processes in large-scale data-based models of laminar cortical circuits

**Background:**
Improving behaviour in response to delayed feedback in new settings is challenging. It is difficult to ascertain the precise cause of errors when feedback on task performance may come seconds, minutes, hours, or even days after the initial source of the error. Additionally, this problem is amplified when one is trying to learn many new tasks sequentially while preserving performance in previously learned tasks.

The success of deep recurrent neural networks (RNNs) in sequential learning tasks has led many neuroscience researchers to ask how the ingredients contributing to their success can be replicated or approximated in biological RNNs [1], [2]. In particular, computational neuroscientists have proposed how components of cortical microcircuits may map onto the gating mechanisms of modern RNNs such as LSTMs and GRUs [3] and how biological neurons can adjust their synapses according to gradient descent via an algorithm approximating backpropagation-through-time (BPTT) [4]–[7]. An underlying assumption of such approaches is that synaptic weights in biological RNNs are available for updates at all times, i.e., fully flexible, and that there should be high-dimensional mechanisms for synapse-by-synapse credit assignment calculations.

At the other extreme, neuroscientists in the recent past often studied RNNs that are completely inflexible, i.e., possess fixed weights. Such networks are known as Echo State Networks (ESNs), or Liquid State Machines when considering their spiking analogues

[8]–[12]. ESNs are defined as having fixed, sparse, random weights that place the network at an edge-of-chaos criticality [11], [12]. This means the unclamped dynamics of such networks are *almost* chaotic, i.e., possess subcritical dynamics. This theoretically imbues the network with a rich set of basis functions that can be mapped with a single set of read-in and read-out layers onto any possible function[13], [14]. Indeed it is possible to train such layers to harness ESN dynamics to perform functions critical for cognition [11], [12].

**Challenge or Controversy**

The theoretical properties of ESNs have led many neuroscientists to look for evidence of subcritical dynamics in neural activity [15]. However, ESNs have had nowhere near the success of LSTM and GRU networks trained with BPTT in achieving benchmark performance in classic machine learning tasks, leading some researchers to question the extent to which ESNs are plausible models of biological RNNs. Furthermore, there is a plethora of evidence that synapses in biological RNNs are plastic [16]–[18] and that it is necessary for improving behaviour [17], [18] and so in the purest sense ESNs cannot account for all learning in real brains. Even so, a closer look at experimental data on synaptic plasticity suggests that at any moment only a fraction of synaptic connections between pyramidal cells is plastic: one finds in any raw data plot from STDP experiments that the weights of a fair number of synapses change very little in response to the STDP induction protocol. Furthermore, experimental data on memory engrams suggest that there exists at any moment only a subset of neurons that are ready to become part of a memory engram [19]. Together, these indicate that learning new tasks requires changing only a small subset of synapses.

Additionally, a notable challenge of research in deep learning is the ability to sequentially train RNNs on new tasks. This is often called *continual learning* in machine learning research, although when talking about biological agents, it might be more appropriate to refer to this as *lifetime learning* since animals will typically have to adapt and re-adapt to new tasks and circumstances throughout their lifetime. It is difficult to achieve good, lasting performance in such a setting with deep neural networks, including RNNs, as weight changes instigated by minimizing cost functions on new tasks interfere with earlier weight changes needed to optimize performance on earlier tasks. This means that performance improvements on earlier tasks suffer when weights are transferred to a new task domain and are allowed to change freely. This scenario is often referred to as the problem of *catastrophic forgetting* [20].

ESNs potentially present a solution to this problem by fixing recurrent networks containing all the expressivity needed to map to new functions, and limiting learning to new read-in/ read-out layers. This could be viewed as a form of transfer learning, where the algorithm for finding a set of weights that provide a universal set of basis functions can be arbitrary in a practical context (e.g., gradient descent, evolutionary algorithms), but in answering questions of biological plausibility the algorithm choice is crucial. Given that biological neural networks must constrain plasticity in order to avoid pathological problems, such as epilepsy [21], [22], it is not entirely unrealistic to assume that, in many networks in the brain, recurrent plasticity is highly constrained and read-out connections are relied on for learning, akin to ESNs.

**Competing hypotheses and proposed approach for resolution:**
The key issue of contention in the debate is how high-dimensional credit assignment signals are likely to be in biological RNNs, and how plastic RNNs in vivo truly are. Is recurrent learning best described as approximating high-dimensional gradient descent through time, or is it possible to achieve high performance across multiple domains with very limited credit information and relatively inflexible recurrent weights?

Multiple routes of inquiry are required to converge on a resolution about the dimensionality of credit signals and degree of plasticity in biological RNNs. Pure machine learning research will continue to assess the potential learning power of artificial RNNs with varying degrees of freedom on weight updates. In particular, developments in novel processor architectures optimized for compiling sparse graphs [23] may lead to new insights into how RNNs with heavily reduced flexibility can be optimized for task-generality.

In parallel, computational neuroscientists should strive to develop models for estimating plasticity amongst unobservable synapses from experimental data, and to identify population-level characteristics of biological RNNs with varying degrees of flexibility that can be validated with experimental methods. Finally, since the prevailing hypothesis regarding synaptic plasticity in biological RNNs stem from variants of reward-modulated three-factor learning rules with very limited flexibility, computational neuroscientists who support hypotheses with greater flexibility should formalize models with end-to-end training of both the task-solving network and the auxiliary credit-assignment system, such as those proposed within a meta-learned 'outer-loop' training framework e.g. [24], [25].

**Concrete outcomes:**
- Establishment and use of continual and lifetime learning benchmarks to assess neural network architectures and training regimes
- Assessment of how performance of ESNs scales with network size
- Determine whether subcritical dynamics emerge in recurrent neural networks trained for lifetime learning.
- Theory providing experimental predictions of network level observations consistent with varying degrees of dimensionality or flexibility in learning.
- Biologically plausible 'outer-loop' learning that moves beyond standard three-factor synaptic plasticity rules.

**Benefit to the community:**
Answering these questions is key to the CCN community for a number of reasons beyond the scientific knowledge itself. First, overcoming challenges in balancing credit-assignment and task-generalization in RNNs is of enormous practical use in developing nonlinear models of sequential data across domains. Second, our work will benefit experimental neuroscientists by proposing high-impact experiments that will be capable of testing our theories and this debate. Third, new descriptive and normative theories generated by our collaboration will inspire new ways to think about links between synaptic plasticity and neuromodulator function.

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
