# OpenReview forum: "Dimensionality and flexibility of learning in biological recurrent neural networks"
_ccneuro.org/CCN/2020/Workshop/GAC_

### Official Review · ~Abraham_Nunes1 · 2020-08-26
**Review of "Dimensionality and flexibility of learning in biological recurrent neural networks"**

**Rating:** 8
**Soundness:** Agree
**Confidence:** 3

**Review:**

Richards et al. pose an interesting question concerning credit assignment in biological neural networks.  The fundamental question relates to the *effective* dimensionality and fixity of synaptic weight updates in biological recurrent networks. Full-rank updates (e.g. where all synapses are available for updating) are characteristic of some recurrent neural network (RNN) architectures that can solve complex problems with impressive performance. Yet, they fail to maintain stability across task domains, which is characteristic of biological neural networks. Conversely, echo state networks (ESNs), which have a reservoir defined by a fixed-weight neural network, may demonstrate greater flexibility across tasks while unfortunately not performing as well as RNNs on classical ML paradigms. However, ESN's have weight updates of effectively much lower dimension than an RNN (i.e. fewer synapses are available for update). The characteristic difference is ostensibly the dimensionality of the objective function domain under each architecture. For an RNN and ESN with the same number of weights, the ESN updates are lower dimensional than those of the RNN since more weights are fixed in the former.

Overall, I found this proposal very well written, and I must applaud the authors for communicating the mathematical ideas so well using only plain language. The authors propose to address an interesting and important problem that will ostensibly benefit the community on both theoretical and practical grounds.

Please see my comments in the next section for more specific details regarding concerns/comments/questions. The proposal's chief limitations surround the concrete objectives, some of which were not clearly compelled by the main body of text. Furthermore, the biologically relevant aspects of this proposal could be better represented in the concrete objectives; I got a sense that the authors aimed mainly to resolve this debate on computational/theoretical grounds, without any clear path to maintain connection to the underlying biology.


**Comments:**

The authors may or may not want to consider some of the following questions/comments/suggestions:

1.	Providing formal definition of the dilemma would facilitate a deeper understanding and motivate specific testable questions.
2.	One thing I found lacking was discussion of the possibility that some of the stability in synaptic weights across tasks is contributed by correlation (in a loose sense of that term) between loss functions/task domains. If there is some shared or roughly invariant structure in the loss function landscapes across tasks, then some weights may remain stable if they encode information related to these shared or invariant properties. I would like to see some discussion about this, but mostly it would be interesting to consider how to test this concretely in the benchmark set.
3.	The establishment of benchmarks is well stated and compelled by the main body. I would be particularly interested in seeing benchmarks that are well-motivated, very simple, but non-trivial, as are commonly used for optimization benchmarking. Will these benchmarks solely be for computational models, or can we develop benchmarks that will also be useful for interrogation of biological neural networks? This would seem necessary for evaluating biological plausibility.
4.	The main body does not clearly compel the ESN size experiments as they would pertain to the central question. I can imagine why this would be interesting in general, but I’m not totally clear on why it is interesting in this specific debate.
5.	I’m not clear on whether the outcome concerning determination of whether RNNs exhibit subcritical dynamics is exploratory, or whether there is some specific reason to believe that they should. It would be helpful to have some more motivation of this.
6.	The authors might consider rephrasing “Biologically plausible ‘outer-loop’ learning that moves beyond standard three-factor synaptic plasticity rules” to more clearly state an objective.
7.	The concrete objectives seem highly theoretical, at the expense of the biological side of this proposal. It seems necessary to have some concrete objective identifying how both sides of this debate can or will be compared on biological grounds. This would ideally move beyond the idea that ESNs are worse models of biological networks if they cannot outperform RNNs on ML tasks. Rather, it would be more useful to have a discussion of which biological properties/data should be explained by the competing architectures in order to resolve this debate.

**Controversy:**

Strongly agree

**Definition:**

Agree

**Expertise:**

Strongly agree

**Outcomes:**

Agree

---

> ### Public Comment · ~Luke_Yuri_Prince1 · 2020-09-07
> **Author reply: part 1**
>
> ***The authors may or may not want to consider some of the following questions/comments/suggestions:***
>
> ***1. Providing a formal definition of the dilemma would facilitate a deeper understanding and motivate specific testable questions.***
>
> A more precise definition of the dilemma is as follows: learning and stability are two dimensions to the behaviour and performance of recurrent neural networks (both artificial and biological) that are seemingly difficult to reconcile. Our dilemma is in determining to what extent stability in biological neural networks will be solved by optimizing over the correct objective function, or whether there are fundamental constraints in the brain and the problem of lifetime learning that ensure many synaptic weights remain relatively unchanged throughout our lifetime. We can formalize this problem mathematically in our position paper.
>
> ***2. One thing I found lacking was discussion of the possibility that some of the stability in synaptic weights across tasks is contributed by correlation (in a loose sense of that term) between loss functions/task domains. If there is some shared or roughly invariant structure in the loss function landscapes across tasks, then some weights may remain stable if they encode information related to these shared or invariant properties. I would like to see some discussion about this, but mostly it would be interesting to consider how to test this concretely in the benchmark set.***
>
> This is a very interesting addition to our proposal and concerns related questions about how optimizing multiple coexisting cost functions can drive synaptic weight changes despite their potentially adversarial nature. Another perspective is that certain objectives may in fact be complementary or cooperative and so weights may be encouraged to remain stable that can, as you say, encode information across multiple domains. All authors on this proposal will likely have diverging opinions about how such objectives interact and their consequences for representations learned by recurrent neural networks. As such, we would be very happy to dedicate part of our debate/ workshop to discussing these issues.
>
> ***3. The establishment of benchmarks is well stated and compelled by the main body. I would be particularly interested in seeing benchmarks that are well-motivated, very simple, but non-trivial, as are commonly used for optimization benchmarking. Will these benchmarks solely be for computational models, or can we develop benchmarks that will also be useful for interrogation of biological neural networks? This would seem necessary for evaluating biological plausibility.***
>
> These benchmarks will solely be for computational models initially, and will be an important filter for proposing models that can be compared to experimental data. We argue that a necessary component of modern computational models of biological learning should be that they can achieve human-comparable performance on cognitively demanding tasks. Given the challenges with proposing competing models that can pass this filter, we feel it is somewhat premature to consider how to interrogate biological neural networks with related benchmarks. However we are aware of efforts by behavioural and computational neuroscientists to develop analogues of common ML benchmark tasks for testing rodent performance, which would be very important for comparison with computational models. Although this is outside our expertise, we would be keen to hear from researchers contributing to these efforts.
>
> ***4. The main body does not clearly compel the ESN size experiments as they would pertain to the central question. I can imagine why this would be interesting in general, but I’m not totally clear on why it is interesting in this specific debate.***
>
> To clarify, it is unknown whether the apparent limitations of ESNs are simply due to insufficient computational resources in creating models of sizes comparable to biological recurrent networks such as the CA3 hippocampal network. The viability of ESNs as a model of learning in biological recurrent networks rests on them being capable of achieving human comparable performance on cognitively demanding tasks. We motivated this debate by considering the extremes: ESNs with no flexibility in recurrent weights, and RNNs trained with BPTT that have complete flexibility in recurrent weights. There is a vast spectrum of middle ground positions here, but it would potentially be premature to rule out this extreme case. We can make this clearer in our revisions.

---

> > ### Public Comment · ~Luke_Yuri_Prince1 · 2020-09-07
> > **Author reply part 2**
> >
> > ***5. I’m not clear on whether the outcome concerning determination of whether RNNs exhibit subcritical dynamics is exploratory, or whether there is some specific reason to believe that they should. It would be helpful to have some more motivation of this.***
> >
> > The reason for testing whether RNNs trained to successfully perform continual learning tasks might exhibit subcritical dynamics is motivated by the observation that Fisher Information maximisation is a useful objective in finding weights for both ESNs and RNNs trained with BPTT. This may lead to a bridge between these two extreme positions that could motivate insights into apparent subcritical dynamics observed in biological RNNs. We can expand on this motivation in our revision.
> >
> > ***6. The authors might consider rephrasing “Biologically plausible ‘outer-loop’ learning that moves beyond standard three-factor synaptic plasticity rules” to more clearly state an objective.***
> >
> > The concrete objective is to propose models with high-dimensional neuromodulatory signals (the third of three factors). Typical models with three factors consider global, scalar quantities modelling neuromodulators that do not consider the complexity of neuromodulatory networks in their own right. Such high-dimensional neuromodulatory networks are perceived as a candidate for implementing biologically plausible lifetime-learning, however as of yet there are few proposed models that have achieved strong benchmark performance. We can highlight this  in our revision
> >
> > ***7. The concrete objectives seem highly theoretical, at the expense of the biological side of this proposal. It seems necessary to have some concrete objective identifying how both sides of this debate can or will be compared on biological grounds. This would ideally move beyond the idea that ESNs are worse models of biological networks if they cannot outperform RNNs on ML tasks. Rather, it would be more useful to have a discussion of which biological properties/data should be explained by the competing architectures in order to resolve this debate.***
> >
> > We agree that ultimately it is necessary to compare theoretical models to experimental data. Nevertheless as stated previously, we contend that it is a significant hurdle in its own right to propose biologically plausible models that can achieve strong performance. Nevertheless we are keen for experimental neuroscientists to take an interest in this debate who can offer methods in which these models can be tested.

---

### Official Review · ~Jenia_Jitsev1 · 2020-08-26
**Essense of Recurrent Processing for Learning**

**Rating:** 6
**Soundness:** Neutral
**Confidence:** 4

**Review:**

The proposed workshop aims to address an important topic in computational neuroscience and machine learning - the significance of recurrent connections and forms of their plasticity in both biological and artificial neural networks. Thereby, authors stress the challenge to study and replicate the ability of the networks to cope with learning long range dependencies hidden in multiple tasks from data that continually streams in through the sensory input.

While I enjoyed reading the proposal and found enough food for thoughts in it to enable fruitful discussion on controversies bound to that topic (for instance, the form and amount of plasticity necessary for circuits to operate in desired way), my impression was that the proposal was losing its focus as it progressed and has general issues with defining one particular clear direction to address, becoming more and more a grand attempt to solve almost everything in one package.

One thing that became apparent to me is that for instance the initial take on recurrent connectivity was vanishing in the background while narration progressed, being replaced by many rather generic questions that can be summarized as a quest to understand and enable efficient continual meta-learning that copes well with both long-range dependencies and existence of multiple tasks. While this is something we all would like to achieve, in the particular proposal it leads to extreme broadening of range of questions to be addressed.

Authors make one clear case of controversy out of the notion that some models see all weights in the network as ever changing parameters, while others - and here they cite reservoir computing like liquid state machines (LSMs) and echo state networks (ESNs) - keep a large fraction of weights frozen. From this, they derive an agenda to study the amount of "raw" plasticity in the biological networks, and how this amount may reflect functional properties of the networks. To me, this motivation seems somewhat hand wavy, for two reasons.

One reason is that good performance on rather simple tasks was observed in many studies that featured randomly initialized networks or some sort of random projection from the input, which did not involve LSMs or ESNs but rather conventional deep convolutional nets (e.g for small selection, http://openaccess.thecvf.com/content_cvpr_2018/papers/Ulyanov_Deep_Image_Prior_CVPR_2018_paper.pdf, https://openreview.net/forum?id=rJNwDjAqYX, https://papers.nips.cc/paper/8777-weight-agnostic-neural-networks) Those studies were also stating that while surprisingly good performance was obtained on data very close to training set, when it came to generalization performance, networks with randomly initialized weights were performing much worse than those with weights adapted during training. Given that observation, it becomes conceivable that performance observed with fixed LSM or ESN reservoirs may be restricted just to those rather simple toy tasks that are still used to train them. It is therefore probably not a good idea to overgeneralize the observed performance and make a claim that fixed random weights reservoirs are good for performing on multitude of scenarios within same network.

Another reason is that susceptibility of a weight for modification in a biological network is not something graved in stone from the beginning of circuit/network existence, but rather itself a dynamical entity controlled by different factors. So the amount of plastic synapses can change depending on given situation and history of previous modifications. It is for instance known that neuromodulation via different transmitters like acetylcholine, dopamine, serotonine, noradrenaline (following from old classical work by Wolf Singer in 80s and many others), other slower acting substances like neuropeptides or hormones, local circuit state itself, etc may gate plasticity within circuits in a way that it could be completely disabled under certain conditions, or enable under others. In general, meta-plasticity mechanisms (e.g sliding LTP/LTD thresholds, used in rules like BCM or ABS, or plasticity of intrinsic excitability) may control the extent to which degree synapses may undergo changes. This evidence draws a picture of a synapse as a rather very complex computational unit with different hidden local synaptic states (see also cascade synapse models by Fusi, Clopath, etc), which gives a very different take on the debate about gating plasticity that is not focusing on role of recurrent connectivity.

My further impression with regard to actually intended focus on role of recurrent connectivity is that it is not becoming clear enough what aspects of it should be studied in the proposal. Most of the current network models incorporating recurrent connectivity mean rather local recurrent connections, which would probably correspond to excitatory or inhibitory recurrent connections on the scale of a cortical microcircuit. This is the case for both LSTM, GRU and also LSMs and ESNs reservoirs (although functional motivation for those models is very different - fighting instable gradients with gating in LSTM like cells, or shaping reservoir dynamics towards somewhat reasonable regime in LSMs). If authors intend to deal only with significance and role of local recurrent connections of that scale, it has to be made more clear. As for sure, role of long-range recurrent connectivity, both of cortical-cortical or subcortical-cortical kind, is essential and hardly understood in terms of information processing and learning.

There is also clear dominance of LSMs as a model of choice. In my opinion it would make proposal more valuable if other quite prominent models of recurrent computation would enter discussion, e.g winner-take-all like circuits, that also had substantial support from neuroscience side (Douglas et al, Science, 1995, Douglas & Martin, Ann Rev Neurosci, 2004), or winnerless competition that is in some sense similar in spirit to reservoir dynamics in LSMs (Rabinovich et al, PLoS Comp Bio, 2008)

Another aspect that is I think under-expressed in the proposal but would be important for its impact and profile sharpening, is the discussion on types of losses used for learning in biological and artificial neural nets. At least the distinction between local losses and global losses seems to me important in the context here. Global losses used in most conventional feed-forward networks implicitly do contain a recurrent connectivity part, as signal conveyed by backpropagation has to travel through the network somehow. Local losses make this assumption explicit, as the basis for computing local losses are signals coming from different parts of the network. The type and flavor of those local losses, e.g predictive coding based, which may be also more task-agnostic that global losses, thus offering better intrinsic support for meta-learning, determine how local circuitry and larger network organization has to look like to support the certain form of necessary computation induced by the utilized loss type. The discussion about type of error signal, or credit signal, necessary for learning in the circuits could be also derived from the assumptions about loss type. In this sense, talking explicitly about loss types and what type of learning those should drive may provide more principled way to talk about what role recurrent connectivity may take in that.

**Comments:**

Given the preceeding discussion, in my opinion it would strongly increase the value and sharpness of the proposal if the authors would re-focus it on role of processing and learning that is critically enabled by recurrent connectivity as opposed to feed-forward networks. The motivation to study recurrent connectivity should be expressed stronger in terms of function necessarily provided by recurrent connectivity for learning. For instance, starting from notion of local vs global loss driven learning may motivate necessity to use different biologically plausible mechanisms to be able to work with local losses, and one of such mechanisms supporting suitable computations may turn out to be recurrent connections. Maybe it would be also good for sake of clarity to treat independent things like enabling meta-learning via complex synapse models separately (as apart from hidden synaptic states, we have short term synaptic depression and facilitation rendering synapses dynamical entities, which makes the whole story too expansive to put in one focused workshop package)

As for the concrete outcomes section at the end of the proposal, following comments:

- "Establishment and use of continual and lifetime learning benchmarks to assess
neural network architectures and training regimes" ---> Such benchmarks are already out there, see for instance ProcGen, Meta World, POET, and can be adapted for test purpose (see eg. Meta-World: A Benchmark and Evaluation for Multi-Task and Meta Reinforcement Learning, https://arxiv.org/abs/1910.10897, https://github.com/rlworkgroup/metaworld, https://meta-world.github.io/; https://openai.com/blog/procgen-benchmark/;)
- "Theory providing experimental predictions of network level observations consistent with varying degrees of dimensionality or flexibility in learning." ---> This seems to me very ambitious given the focus on rather specific, limited architectures in the proposal that do not yet take into account many basic properties of cortical organization and processing
- "Biologically plausible ‘outer-loop’ learning that moves beyond standard three-factor
synaptic plasticity rules."
---> There is already a pletora of meta-plasticity learning rules beyond three factor standard ones, for instance BCM or ABS that use LTP/LTD sliding threshold updated by previous experience (Bienenstock et al, J Neurosci, 1982; Artola et al, Nature, 1990; Abraham & Bear, Trends Neurosci, 1996), or families of homeostatic and intrinsic plasticity rules (Desai et al, Nature Neurosci, 1999; Turrigiano, Ann Rev Neurosci 2011), or cascade synapse models with hidden variables (https://www.nature.com/articles/nn.4401.pdf, https://papers.nips.cc/paper/4872-a-memory-frontier-for-complex-synapses, http://proceedings.mlr.press/v97/kaplanis19a.html). It should be discussed which of those can be already taken as basis for achieving intended meta-learning functionality.

**Controversy:**

Agree

**Definition:**

Neutral

**Expertise:**

Strongly agree

**Outcomes:**

Neutral

---

> ### Public Comment · ~Luke_Yuri_Prince1 · 2020-09-07
> **Author reply: part 1**
>
> ***My impression was that the proposal was losing its focus as it progressed and has general issues with defining one particular clear direction to address.***
>
> We agree that this is a very broad topic. In our proposal, we have attempted to remain focused by motivating the debate at the extremes (ESNs with no flexibility vs RNNs trained with BPTT with complete flexibility). Clearly there is a huge middle ground in positions, and many of the coauthors on this proposal would identify with such middle ground positions. We think this apparent lack of focus was caused by not offering sufficient motivation for this middle ground in the proposal and perhaps focussed too strongly on these extremes. In our revision we will incorporate greater discussion of the spectrum of models that lie between these extremes. We can also clarify how our research objectives fit more concretely with this spectrum once we have outlined it.
>
> ***The initial take on recurrent connectivity was vanishing in the background while narration progressed, being replaced by many rather generic questions that can be summarized as a quest to understand and enable efficient continual meta-learning that copes well with both long-range dependencies and existence of multiple tasks.***
>
> We will rephrase the problem in the vein of our response to another reviewer, and hopefully this will convince you that we have a strong focus on recurrent networks. The source of controversy we wish to address is to determine the extent to which stability and flexibility in recurrent neural networks can be reconciled. There may be fundamental constraints to the problem of lifetime learning that keeps many recurrent weights staying relatively fixed throughout life and that evolutionary mechanisms have managed to work with these constraints to balance generalization (improved performance across multiple tasks) and specialisation (forgetting past tasks to optimize current tasks). In contrast, others argue that we simply have not yet found the right objective function that controls when weights remain fixed or are free to change.
>
> We would argue that continual meta-learning is the crucial framework offered by modern artificial intelligence research for understanding biological recurrent neural networks. Tasks are both ill-defined and sequential, necessitating self- or unsupervised learning of objectives and feedback. We will attempt to motivate why we think this is a crucial and logical step for framing this debate in our revision.
>
> ***Authors make one clear case of controversy out of the notion that some models see all weights in the network as ever changing parameters, while others - and here they cite reservoir computing like liquid state machines (LSMs) and echo state networks (ESNs) - keep a large fraction of weights frozen. From this, they derive an agenda to study the amount of "raw" plasticity in the biological networks, and how this amount may reflect functional properties of the networks. To me, this motivation seems somewhat hand wavy, for two reasons.
> One reason is that good performance on rather simple tasks was observed in many studies that featured randomly initialized networks or some sort of random projection from the input, which did not involve LSMs or ESNs but rather conventional deep convolutional nets. Those studies were also stating that while surprisingly good performance was obtained on data very close to training set, when it came to generalization performance, networks with randomly initialized weights were performing much worse than those with weights adapted during training. Given that observation, it becomes conceivable that performance observed with fixed LSM or ESN reservoirs may be restricted just to those rather simple toy tasks that are still used to train them. It is therefore probably not a good idea to overgeneralize the observed performance and make a claim that fixed random weights reservoirs are good for performing on multitude of scenarios within same network.***
>
> These comments about the generalization performance of randomly initialized deep convolutional networks are hugely relevant and are key fodder in this debate. For authors believing in full-rank temporal credit assignment in biological recurrent neural networks, this is an additional reason to discount the usefulness of fixed networks. We had not considered this angle when writing this proposal, and are thankful to the reviewer for raising this.

---

> > ### Public Comment · ~Luke_Yuri_Prince1 · 2020-09-07
> > **Author reply: part 2**
> >
> > ***Another reason is that susceptibility of a weight for modification in a biological network is not something graved in stone from the beginning of circuit/network existence, but rather itself a dynamical entity controlled by different factors. So the amount of plastic synapses can change depending on given situation and history of previous modifications. It is for instance known that neuromodulation via different transmitters like acetylcholine, dopamine, serotonine, noradrenaline (following from old classical work by Wolf Singer in 80s and many others), other slower acting substances like neuropeptides or hormones, local circuit state itself, etc may gate plasticity within circuits in a way that it could be completely disabled under certain conditions, or enable under others. In general, meta-plasticity mechanisms (e.g sliding LTP/LTD thresholds, used in rules like BCM or ABS, or plasticity of intrinsic excitability) may control the extent to which degree synapses may undergo changes. This evidence draws a picture of a synapse as a rather very complex computational unit with different hidden local synaptic states (see also cascade synapse models by Fusi, Clopath, etc), which gives a very different take on the debate about gating plasticity that is not focusing on role of recurrent connectivity.***
> >
> > Indeed, the reviewer raises a valid point and offers multiple candidates for considering routes to consider how biologically plausible high-dimensional credit assignment could be achieved. Indeed, neural network models typically consider synapses and neuromodulatory inputs as scalar quantities, when they are in fact dynamic, multi-dimensional vector quantities in their own right. We would be happy to include a discussion of this in our workshop.
> >
> > ***My further impression with regard to actually intended focus on role of recurrent connectivity is that it is not becoming clear enough what aspects of it should be studied in the proposal. Most of the current network models incorporating recurrent connectivity mean rather local recurrent connections, which would probably correspond to excitatory or inhibitory recurrent connections on the scale of a cortical microcircuit. This is the case for both LSTM, GRU and also LSMs and ESNs reservoirs (although functional motivation for those models is very different - fighting instable gradients with gating in LSTM like cells, or shaping reservoir dynamics towards somewhat reasonable regime in LSMs). If authors intend to deal only with significance and role of local recurrent connections of that scale, it has to be made more clear. As for sure, role of long-range recurrent connectivity, both of cortical-cortical or subcortical-cortical kind, is essential and hardly understood in terms of information processing and learning.***
> >
> > We did not go into much detail on the constraints on connectivity being proposed in our model other than rough magnitudes of connection probabilities expected in biological networks. We are aware of research into how the structure of such graphs can be important for information processing and learning, however we felt this was something of a diversion from the key aim of the topic. Indeed, if the reviewer’s main concern is that they would like us to be more focussed, we think we can safely and respectfully say that this is not a topic that would feature too heavily for us.
> >
> > ***There is also clear dominance of LSMs as a model of choice. In my opinion it would make proposal more valuable if other quite prominent models of recurrent computation would enter discussion, e.g winner-take-all like circuits, that also had substantial support from neuroscience side (Douglas et al, Science, 1995, Douglas & Martin, Ann Rev Neurosci, 2004), or winnerless competition that is in some sense similar in spirit to reservoir dynamics in LSMs (Rabinovich et al, PLoS Comp Bio, 2008)***
> >
> > As we have mentioned in our other responses, we acknowledge there is a spectrum of models dealing with recurrent computation with varying degrees of flexibility. However, we chose to motivate this debate at the extremes. We can discuss this middle ground in our revisions.

---

> > > ### Public Comment · ~Luke_Yuri_Prince1 · 2020-09-07
> > > **Author reply: part 3**
> > >
> > > ***Another aspect that is I think under-expressed in the proposal but would be important for its impact and profile sharpening, is the discussion on types of losses used for learning in biological and artificial neural nets. At least the distinction between local losses and global losses seems to me important in the context here. Global losses used in most conventional feed-forward networks implicitly do contain a recurrent connectivity part, as signal conveyed by backpropagation has to travel through the network somehow. Local losses make this assumption explicit, as the basis for computing local losses are signals coming from different parts of the network. The type and flavor of those local losses, e.g predictive coding based, which may be also more task-agnostic that global losses, thus offering better intrinsic support for meta-learning, determine how local circuitry and larger network organization has to look like to support the certain form of necessary computation induced by the utilized loss type. The discussion about type of error signal, or credit signal, necessary for learning in the circuits could be also derived from the assumptions about loss type. In this sense, talking explicitly about loss types and what type of learning those should drive may provide more principled way to talk about what role recurrent connectivity may take in that.***
> > >
> > > As mentioned to another reviewer, we think this is a hugely interesting topic for this debate that we would like to incorporate. The authors of this proposal certainly have diverging opinions on the importance of global vs local cost functions that are of a potential adversarial or cooperative nature.
> > >
> > > ***Given the preceeding discussion, in my opinion it would strongly increase the value and sharpness of the proposal if the authors would re-focus it on role of processing and learning that is critically enabled by recurrent connectivity as opposed to feed-forward networks. The motivation to study recurrent connectivity should be expressed stronger in terms of function necessarily provided by recurrent connectivity for learning. For instance, starting from notion of local vs global loss driven learning may motivate necessity to use different biologically plausible mechanisms to be able to work with local losses, and one of such mechanisms supporting suitable computations may turn out to be recurrent connections. ***
> > >
> > > We disagree that we should pivot the focus of the proposal towards a debate on global vs local loss functions. While the debate is relevant to ours, as discussed with another reviewer, we maintain that the core issue of flexibility of learning in recurrent networks merits debate on its own. This does not diminish the fact that the relative importance or even existence of global vs local loss functions are a distinct source of controversy in their own right in neuroscience worthy of their own GAC proposal!
> > >
> > > ***Maybe it would be also good for sake of clarity to treat independent things like enabling meta-learning via complex synapse models separately (as apart from hidden synaptic states, we have short term synaptic depression and facilitation rendering synapses dynamical entities, which makes the whole story too expansive to put in one focused workshop package)***
> > >
> > > As the reviewer notes further down, there are many complex synapse models that may be candidates for analytical treatment as high-dimensional temporal credit assignment mechanisms. We can strengthen the case for motivating the debate and ensuing model development within a biologically plausible continual meta-learning framework in our revisions.

---

> > > > ### Public Comment · ~Luke_Yuri_Prince1 · 2020-09-07
> > > > **Author reply: part 4**
> > > >
> > > > ***"Establishment and use of continual and lifetime learning benchmarks to assess neural network architectures and training regimes" ---> Such benchmarks are already out there, see for instance ProcGen, Meta World, POET, and can be adapted for test purpose (see eg. Meta-World: A Benchmark and Evaluation for Multi-Task and Meta Reinforcement Learning, https://arxiv.org/abs/1910.10897, https://github.com/rlworkgroup/metaworld, https://meta-world.github.io/; https://openai.com/blog/procgen-benchmark/;)***
> > > >
> > > > Thank you for highlighting existing benchmarks. We will cite these in our revisions.
> > > >
> > > > ***"Theory providing experimental predictions of network level observations consistent with varying degrees of dimensionality or flexibility in learning." ---> This seems to me very ambitious given the focus on rather specific, limited architectures in the proposal that do not yet take into account many basic properties of cortical organization and processing***
> > > >
> > > > This aim is perhaps too imprecise and ambitious. Since the authors are all primarily theoretical neuroscientists, we can reach out to experimental collaborators willing to offer their expertise in developing experimental predictions from our debate, and potentially devote a topic in our workshop to it.
> > > >
> > > > ***"Biologically plausible ‘outer-loop’ learning that moves beyond standard three-factor synaptic plasticity rules."
> > > > --> There is already a pletora of meta-plasticity learning rules beyond three factor standard ones, for instance BCM or ABS that use LTP/LTD sliding threshold updated by previous experience (Bienenstock et al, J Neurosci, 1982; Artola et al, Nature, 1990; Abraham & Bear, Trends Neurosci, 1996), or families of homeostatic and intrinsic plasticity rules (Desai et al, Nature Neurosci, 1999; Turrigiano, Ann Rev Neurosci 2011), or cascade synapse models with hidden variables (https://www.nature.com/articles/nn.4401.pdf, https://papers.nips.cc/paper/4872-a-memory-frontier-for-complex-synapses, http://proceedings.mlr.press/v97/kaplanis19a.html). It should be discussed which of those can be already taken as basis for achieving intended meta-learning functionality.***
> > > >
> > > > We would like to note a semantic distinction between our concepts of ‘learning rules’ and ‘synaptic plasticity rules’. While we are well aware of a myriad of complex synaptic plasticity rules, very few have been treated in the context of being used as greedy update rules in typical machine learning tasks. We make the distinction of ‘learning rules’ as those that have been successfully applied to optimizing performance on a specific set of tasks. Such synaptic plasticity rules are of course great candidates for inspiring such greedy update rules.

---

### Official Review · ~Brian_Cheung1 · 2020-08-26
**Review for "Dimensionality and flexibility of learning in biological recurrent neural networks"**

**Rating:** 7
**Soundness:** Agree
**Confidence:** 3

**Review:**

Backpropagation through time has become the defacto standard for dealing with credit assignment for time series. While the machine learning community has shown this is an effective strategy for existing problems, it faces challenges for future problems. This proposal discusses one key challenge which is catastrophic forgetting and proposes using minimally adapted structures like echo state networks as the foundation of a potential solution. The inability of artificial learning algorithms to stably learn and much less flourish when data is presented in a structured manner remains a glaring discrepancy between their biological counterparts. This proposal is an attempt to resolve this gap and should lead to a better understanding of learning algorithms in both domains. This has the potential to be a significant contribution.

See below for additional comments on pros/cons of the proposal:


**Comments:**

The authors mention the "Establishment and use of continual and lifetime learning benchmarks to assess neural network architectures and training regimes". I agree new benchmarks for continual/lifetime learning need to be established as the current benchmarks are simply concatenations of already existing datasets which were collected without lifetime learning in mind. But what are the difference between what the authors plan to create as a benchmark in this proposal as compared to what already exists. Will the benchmarks be more reflective of biological learning? For instance, will the datasets look something like the CRIB dataset (https://iolfcv.github.io/)?

The authors may also want to consider how their ESN proposal fits in with recent work [1] where the training signal provided to the model is much weaker than standard backpropagation. Similar to ESNs, [1] does not train the network weights but simply learns masks over random weights to alleviate the catastrophic forgetting issue. There is a growing consensus that weights may not be as critical as the pathways of activations that flow through them. Are the authors proposing a learning paradigm where the foundation will still be closely related to backpropagation like these other works? Or will a substantially different learning paradim be necessary. Even if a different learning paradim is not being proposed, will the proposal provide any insight to determine if current learning paradigms like backpropagation is a good foundation to start from? This would give helpful insight to the community even if the proposed ESN is not fully successful in preventing catastrophic forgetting.

[1] "Supermasks in Superposition", https://arxiv.org/abs/2006.14769
[2] "What's Hidden in a Randomly Weighted Neural Network?", https://arxiv.org/abs/1911.13299

**Controversy:**

Agree

**Definition:**

Strongly agree

**Expertise:**

Agree

**Outcomes:**

Agree

---

> ### Public Comment · ~Luke_Yuri_Prince1 · 2020-09-07
> **Author reply**
>
> ***Backpropagation through time has become the defacto standard for dealing with credit assignment for time series. While the machine learning community has shown this is an effective strategy for existing problems, it faces challenges for future problems. This proposal discusses one key challenge which is catastrophic forgetting and proposes using minimally adapted structures like echo state networks as the foundation of a potential solution. The inability of artificial learning algorithms to stably learn and much less flourish when data is presented in a structured manner remains a glaring discrepancy between their biological counterparts. This proposal is an attempt to resolve this gap and should lead to a better understanding of learning algorithms in both domains. This has the potential to be a significant contribution***
>
> While we are grateful to the reviewer for their positive comments on our proposal, we would like to clarify that the extreme case of ESNs is used as a way to motivate and frame the debate. The authors of the proposal are keen to cover a broad spectrum of positions on degrees of flexibility in recurrent neural networks.
>
> ***The authors mention the "Establishment and use of continual and lifetime learning benchmarks to assess neural network architectures and training regimes". I agree new benchmarks for continual/lifetime learning need to be established as the current benchmarks are simply concatenations of already existing datasets which were collected without lifetime learning in mind. But what are the difference between what the authors plan to create as a benchmark in this proposal as compared to what already exists. Will the benchmarks be more reflective of biological learning? For instance, will the datasets look something like the CRIB dataset (https://iolfcv.github.io/)?***
>
> Yes, the idea would be to develop benchmarks with ecological relevance that could lead to comparisons between learning algorithms and human or animal performance through experiments. The CRIB dataset is a good example of this.
>
> ***The authors may also want to consider how their ESN proposal fits in with recent work [1] where the training signal provided to the model is much weaker than standard backpropagation. Similar to ESNs, [1] does not train the network weights but simply learns masks over random weights to alleviate the catastrophic forgetting issue. There is a growing consensus that weights may not be as critical as the pathways of activations that flow through them.
> [1] "Supermasks in Superposition", https://arxiv.org/abs/2006.14769 [2] "What's Hidden in a Randomly Weighted Neural Network?", https://arxiv.org/abs/1911.13299***
>
> This is an excellent question and contribution. As discussed with another reviewer, there is considerable interest in the potential of optimizing network architectures (which we consider synonymous with processing pathways) that can have randomly initialized weights yet still achieve strong performance. As Reviewer 2 noted, in convnets, random initialization does not lead to good generalization. This would be a key topic for our debate.
>
> ***Are the authors proposing a learning paradigm where the foundation will still be closely related to backpropagation like these other works? Or will a substantially different learning paradim be necessary. Even if a different learning paradim is not being proposed, will the proposal provide any insight to determine if current learning paradigms like backpropagation is a good foundation to start from? This would give helpful insight to the community even if the proposed ESN is not fully successful in preventing catastrophic forgetting.***
>
> Some of the authors strongly believe that backpropagation is the best foundation to start from. Nevertheless, it is not the only foundation. We are also keen to discuss methods relying on greedy layer-wise update rules, genetic algorithms, amongst others.